# Data Difficulty and the Generalization–Extrapolation Tradeoff in LLM Fine-Tuning

**Siyuan Liu** [* 1] **Tinghong Chen** [* 2 3] **Xinghan Li** [* 1] **Yifei Wang** [4] **Jingzhao Zhang** [1 3]

## Abstract

Data selection during supervised fine-tuning (SFT) can critically change the behavior of large language models (LLMs). Although existing work has studied the effect of selecting data based on heuristics such as perplexity, difficulty, or length, the reported findings are often inconsistent or context-dependent. In this work, we systematically study the role of data difficulty in fine-tuning from both empirical and theoretical perspectives, and find that there is no universally optimal difficulty level; rather, its effectiveness depends on the dataset size. We show that for a fixed data budget, there exists an optimal data difficulty for SFT, and that this optimal difficulty shifts toward harder data as the data budget increases. To explain this phenomenon, we conduct controlled synthetic experiments that reveal a simple underlying mechanism: the interplay between the (in-distribution) generalization gap and the extrapolation gap. We further support this mechanism through a theoretical analysis using PAC-Bayesian generalization bounds. Overall, our results clarify how data size and difficulty jointly affect the trade-off between generalization and extrapolation in SFT, providing guidance for difficulty-based data selection under certain model and data conditions.

## 1. Introduction

Selecting training data for supervised fine-tuning (SFT) has a large impact on downstream performance of large language models. Among the various heuristics used in practice, **data difficulty**, which reflects both the complexity of a

*Equal contribution [1]Institute for Interdisciplinary Information Sciences, Tsinghua University, China [2]College of Artificial Intelligence, Tsinghua University, China [3]Shanghai Qi Zhi Institute, China [4]Amazon AGI SF Lab, USA. Correspondence to: Siyuan Liu <liusiyua23@mails.tsinghua.edu.cn>, Jingzhao Zhang <jingzhaoz@mail.tsinghua.edu.cn>.

*Proceedings of the 43rd International Conference on Machine Learning*, Seoul, South Korea. PMLR 306, 2026. Copyright 2026 by the author(s).

Table 1. SFT results of Qwen2.5-Math-1.5B (Yang et al., 2024) on easy, medium, and hard subsets of OpenR1-Math-94k (Hugging Face, 2025), OpenMath (Moshkov et al., 2025), and OpenScience (NVIDIA, 2025). We measure difficulty using Chain-of-Thought (CoT) length. Models trained on math datasets are evaluated on Math500, while those trained on the science dataset are evaluated on MMLU.

| DATASET | EASY | MEDIUM | HARD |
|---|---|---|---|
| OPENR1-MATH-94K | 61.1 | **68.3** | 61.7 |
| OPENMATH (200K SUBSET) | **71.7** | 70.1 | 69.0 |
| OPENSCIENCE (200K SUBSET) | **53.4** | 53.0 | 41.2 |

question and the learnability of its solution, is both intuitive and controversial. On one hand, many studies suggest that overly easy data are uninformative and should be removed, and that harder examples are more useful for learning (Marion et al., 2023; Xia et al., 2024a; Muennighoff et al., 2025; Ye et al., 2025). On the other hand, another line of work argues that fine-tuning should stay close to the training distribution of the base model, suggesting that easier training samples may lead to better generalization (Zhang et al., 2025; De la Rosa et al., 2022; Ren et al., 2024; Wu et al., 2025). A related third view suggests that data of intermediate difficulty can yield the best performance (Marion et al., 2023; Qi et al., 2025).

These conflicting findings motivate us to re-examine data difficulty through a series of preliminary experiments across different datasets and evaluation tasks. We split each dataset into three equal subsets based on difficulty and evaluate models fine-tuned on each subset. Typically, data difficulty is measured by metrics such as perplexity, correctness, or trajectory length. Here we measure difficulty using ground-truth Chain-of-Thought (CoT) length, following prior studies (Cheng et al., 2025; Yu et al., 2025), and justify this choice in Figure 1; additional discussion on difficulty metrics is provided in Appendix B.

Table 1 shows that the optimal difficulty for SFT is *not universal*. Even with the same model, the same evaluation tasks, and similar data, the optimal data difficulty varies across different datasets. This observation raises an important open question:

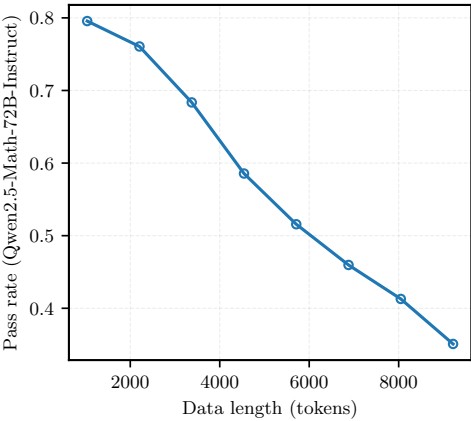

Figure 1. Relationship between data difficulty measured by ground-truth CoT length and pass rate obtained from an external LLM. Results come from the OpenMath dataset, where pass rate is computed from 32 sampled generations using Qwen2.5-Math-72B-Instruct under the Tool-Integrated Reasoning (TIR) mode. Longer CoT lengths are associated with a lower pass rate, indicating higher problem difficulty.

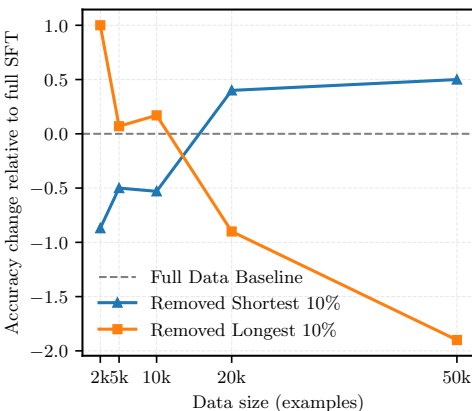

Figure 2. Comparison between SFT with full data and with the shortest or longest 10% of training examples removed. The x-axis corresponds to different training set sizes from OpenR1-Math-94k. The y-axis shows the difference in Math500 accuracy relative to full-data SFT. Removing hard examples helps in the small-data regime, while removing easy examples helps in the large-data regime.

*Under what conditions should SFT prioritize easy, hard, or intermediate-difficulty data?*

Further analysis reveals that many of these conflicting findings can be resolved by considering a previously under-explored factor: **training data size**. Figure 2 shows a notable pattern: removing hard data improves performance on small data scales but degrades it as the data size increases, while removing easy data provides little benefit on small scales but becomes beneficial on larger scales. This finding is consistent with recent observations that large-scale data filtering often offers little advantage over random selection for instruction tuning (Xia et al., 2024b), but existing work has not provided a principled explanation for this behavior.

Motivated by this dependence on data size, we systematically study how the interplay between data size and data difficulty affects SFT performance. Our experiments reveal a clear pattern: **there exists an optimal data difficulty for SFT, which increases as the available training data grows**. This pattern explains the contrasting effects of filtering easy versus hard examples across different data regimes (Figure 2) and provides practical guidance for selecting training data to maximize SFT performance.

To further explain the underlying factors behind this behavior, we conduct a series of controlled synthetic experiments, which reveal that the observed trend arises from the trade-off between two fundamental factors: the (in-distribution) **generalization gap**, namely the performance degradation caused by imperfect generalization within the training distribution, and the **extrapolation gap**, namely the performance degradation induced by the distribution shift between the training data and harder, out-of-distribution test cases. When training data are overly easy, the extrapolation gap dominates and limits extrapolation to challenging test problems; when data are excessively hard, the in-distribution generalization gap becomes dominant, which leads to an overall performance drop, while increasing data size can reduce both gaps. The balance between these two effects gives rise to an optimal data-size-dependent difficulty for supervised fine-tuning. Beyond empirical evidence, we further provide a brief theoretical analysis based on PAC-Bayesian bounds, which establishes an error decomposition into the generalization gap and the extrapolation gap.

Together, our findings provide a novel view of how data difficulty and data scale influence SFT performance. They suggest that effective data selection for supervised fine-tuning should move beyond simply favoring easy or hard data and instead adapt data difficulty to the available data scale and model capability. By connecting empirical phenomena with theoretical analysis, our work advances a principled understanding of how data properties affect learning in large language models and offers guidance for designing training datasets to maximize SFT performance.

**Conflict of Interest Disclosure**   The authors declare no financial conflicts of interest.

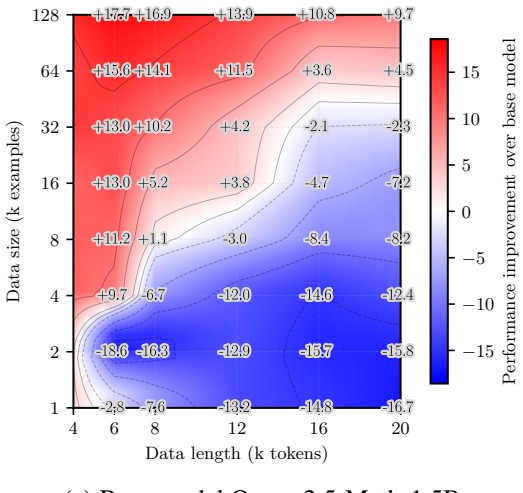

(a) Base model Qwen-2.5-Math-1.5B

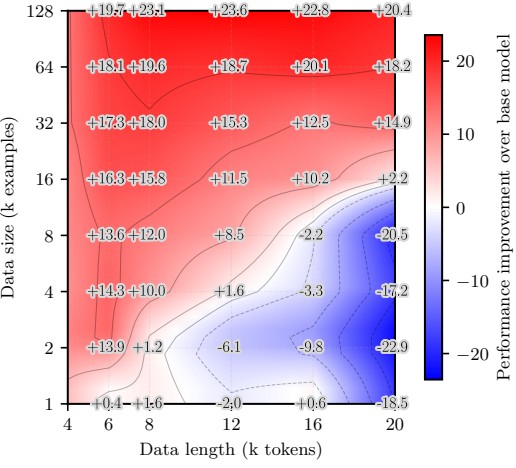

(b) Base model Qwen-2.5-Math-7B

Figure 3. Performance gains over different base models as a function of data size and difficulty, trained on the OpenMath dataset. Performance is evaluated by the average accuracy (%) on Math500, AIME24, and Minerva Math. Data difficulty is measured by ground-truth CoT length.

## 2. Related Work

**Metric-based data selection for supervised fine-tuning.** Some prior work advocates for removing examples that are not informative to the base model. For example, LIMO (Ye et al., 2025) and s1 (Muennighoff et al., 2025) curate data pools by discarding examples that the base model can already solve correctly. LESS (Xia et al., 2024a) selects training examples based on their estimated influence on validation loss.

Another class of methods focuses on data difficulty, using heuristics such as perplexity or response length. BERTIN (De la Rosa et al., 2022) selects examples with intermediate perplexity, where the perplexity is calculated using a fixed $n$-gram language model. Marion et al. (2023) show that pruning data by removing easy instances identified by low perplexity can improve performance. Zhang et al. (2025) show that when multiple responses generated by different LLMs are available for the same instruction, selecting in-distribution responses leads to better fine-tuning performance. Cheng et al. (2025) use CoT length to estimate data difficulty for data selection.

Other approaches rely on token-level reweighting, which implicitly performs difficulty-based selection. Dynamic Fine-Tuning (DFT) (Wu et al., 2025) introduces token-level reweighting that favors high-probability tokens during training. Anchored Supervised Fine-Tuning (Zhu et al., 2025) adds a KL regularization term to the DFT objective to mitigate the distributional drift issue observed in DFT.

**Understanding the role of data in supervised fine-tuning.** Lin et al. (2025) argue that domain-specific SFT can degrade general-domain performance due to an emphasis on low-probability tokens. Li et al. (2025) study likelihood-based objectives, showing that while DFT-like token reweighting can be beneficial for in-distribution data, it often fails to outperform standard SFT under distribution shift. Ren et al. (2024) observe that LLM-generated data are more suitable for SFT than human-annotated data, partly because they exhibit lower perplexity with respect to the base model, thus reducing distribution mismatch.

**Learning-theoretic views on adaptation and distribution shift.** Our analysis is also related to learning-theoretic studies of adaptation, task complexity, and distribution shift. PAC-Bayesian theory provides a standard way to relate generalization to a posterior-prior divergence term, where the KL divergence can be interpreted as a complexity measure relative to a data-independent prior (Alquier, 2024; McAllester, 1998; Maurer, 2004). Information-theoretic analyses further connect task difficulty and transfer to the amount of task-specific information that must be encoded beyond a prior representation (Achille et al., 2021; Harutyunyan et al., 2020; Achille et al., 2019). Separately, domain adaptation analyses often control target-domain risk by combining source-domain generalization with a distribution-shift term, commonly expressed through total variation or KL-based relaxations (Nguyen et al., 2021). Our work uses these tools to formalize an empirical mechanism in SFT: increasing data difficulty can reduce the extrapolation gap to challenging test cases, but may also increase the complexity of adaptation relative to the base model.

Previous studies provide valuable insights into specific factors such as perplexity, token probability, and data source.

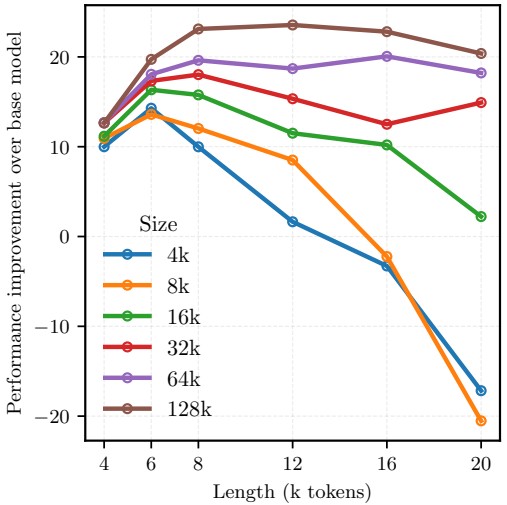

(a) Performance gain as a function of data difficulty with data size held constant.

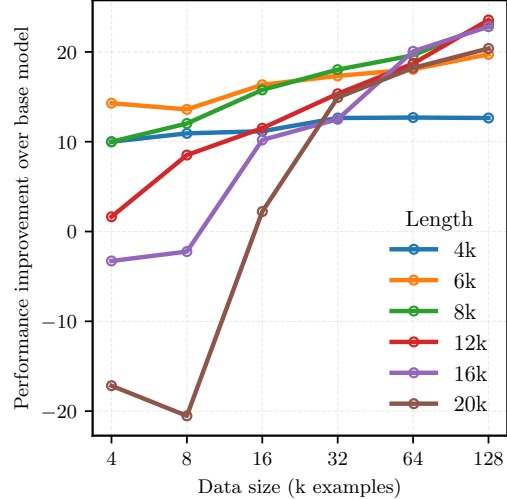

(b) Performance gain as a function of data size with data difficulty held constant.

Figure 4. One-dimensional slices of the 2D data size–difficulty experiment on Qwen-2.5-Math-7B from Section 3.

Our work instead seeks to understand the role of data difficulty by explicitly modeling its interaction with training data size. Existing data selection methods typically rely on fixed selection criteria based on difficulty or informativeness. Our work complements existing data selection approaches by providing a scaling-based understanding of when different types of training data are most effective: easier examples tend to be more effective in low-data regimes, while harder examples become increasingly beneficial as the data budget grows. This helps reconcile seemingly conflicting observations in the literature and provides a principled explanation for them.

## 3. Fine-grained Experiments: The Interplay between Data Difficulty and Size

The effectiveness of filtering hard data depends strongly on the training data size, as we observed in Figure 2. This observation indicates that data difficulty cannot be analyzed in isolation; instead, it is necessary to jointly consider both data size and difficulty to understand their overall influence.

Motivated by this, we perform a systematic two-dimensional study spanning a wide range of data sizes and difficulty levels. Specifically, we conduct SFT on the Qwen2.5-Math-1.5B and Qwen2.5-Math-7B models using the OpenMath dataset, controlling data difficulty by partitioning the training data into disjoint difficulty ranges based on ground-truth CoT length. We then vary data size within each difficulty range. Training details are provided in Appendix A.1. Additional experiments on Llama models and science reasoning tasks are provided in Appendix C.

Figure 3 presents the overall performance landscape as a function of data size and data difficulty. Performance is evaluated by the average accuracy (%) on Math500, AIME24, and Minerva Math to reduce variance, and we report accuracy improvements relative to the pretrained base model. From these results, we observe several clear trends.

**There is an optimal difficulty level for SFT.** As Figure 4(a) shows, for a fixed data size, SFT performance does not change monotonically with difficulty. It first increases as the training data difficulty increases, reaches a clear peak at an intermediate difficulty level, and then drops when the data become too hard. This shows that neither very easy nor very hard data are optimal for SFT and that there exists an optimal difficulty level.

**Performance increases and then saturates as data size grows.** As Figure 4(b) shows, for a fixed difficulty range, SFT performance increases as the training set grows, but the gains gradually slow down and eventually saturate. After a certain point, adding more data no longer leads to meaningful improvements. We also observe that easy data reach saturation earlier than hard data, which is intuitive: easy examples are learned quickly, while harder ones provide more opportunity for improvement.

**Difficulty is relative to base model capability.** As shown in Figure 3, the effectiveness of SFT depends not only on the absolute difficulty of the data but also on the capability of the base model. A stronger base model benefits from a broader range of difficulty levels compared to a weaker model. For the same data size and difficulty range, a stronger model is more likely to learn successfully, while a weaker model may

show limited gains. This suggests that SFT performance is primarily determined by the relative difficulty of the data with respect to the base model's current capability.

The above observations explain the conflicting observations in previous work on how SFT data should be filtered based on difficulty. Next, we aim to identify the underlying factors that drive the change in optimal data difficulty.

# 4. Why Do These Behaviors Arise? Controlled Experiments on iGSM

To explain why the above non-monotonic trends arise, we turn to synthetic reasoning data generated using the iGSM framework (Ye et al., 2024). This framework generates a diverse set of grade-school math reasoning problems in a rule-based manner, allowing precise control over problem difficulty while maintaining consistent data quality and formatting.

## 4.1. Varying Data Size and Difficulty on iGSM Data

**Experiment Setup** In the iGSM problem generator, each problem is generated from an underlying dependency graph, and each vertex represents an arithmetic operation. The number of operations (*ops*) determines the difficulty of each instance, allowing fine-grained control over problem difficulty and reasoning structure. The number of ops is also strongly correlated with CoT length, consistent with our previous difficulty metric.

We use Qwen2.5-Math-1.5B as the base model. To familiarize the model with the task setting, we pretrain the base model on a mid-training dataset consisting of problems whose op-counts are uniformly distributed within a specified range. The resulting model is then treated as the "base model" for subsequent SFT experiments. For simplicity, we denote "Ops[2–8]2k" as the base model mid-trained on data with ops in the range [2,8], using 2,000 samples per op for one epoch.

To evaluate a model's performance, we randomly select 50 samples for each op level in the range $[2, 20]$ to form the test set. We then investigate SFT under varying data sizes and data difficulties (measured by the number of ops). For each experimental configuration, we fine-tune models for three epochs using the same hyperparameters and report performance changes relative to the base model. Additional experimental details are provided in Appendix A.2.

By analyzing the performance landscape across training configurations, we identify several recurring trends in how data difficulty affects supervised fine-tuning performance. We summarize these trends as four key observations below.

---

**Observations from iGSM Experiments**

Figure 5 reveals four consistent observations:

- **Observation 1 (Data size effect).** For a fixed difficulty, SFT performance improves with increasing data size and then saturates.

- **Observation 2 (Non-monotonic difficulty effect).** For a fixed data size, performance first increases then decreases as difficulty increases, exhibiting an interior optimum.

- **Observation 3 (Shifting optimal difficulty).** The optimal difficulty that maximizes performance increases as the available training data size grows.

- **Observation 4 (Model-relative difficulty).** Difficulty is relative to model capability: as the base model becomes stronger, the boundaries [1] and optimal difficulty all shift toward higher difficulty.

---

These findings are consistent with those in Section 3, but are more clearly observed in the controlled iGSM setting. In the following sections, we provide a principled explanation for Observations 1–4.

## 4.2. The Generalization-Extrapolation Tradeoff in SFT

### 4.2.1. DIFFICULTY-DECOMPOSED EVALUATION

To understand the underlying factors behind the above observations, we decompose the test set by difficulty and evaluate performance on each slice independently.

Figure 6 reveals two distinct failure modes of SFT, depending on the data difficulty:

- When SFT is performed on relatively easy data, the model improves on easy test instances but exhibits degraded performance on harder ones, resulting in an overall performance drop. This suggests that the model fits the training distribution well but fails to transfer this improvement to more difficult test cases, indicating a large **train–test mismatch**.

- In contrast, when SFT is performed on very difficult data, the performance degrades uniformly across all test difficulty levels. This behavior indicates that the model fails to effectively learn from the training distribution itself due to insufficient data, leading to poor **in-distribution generalization**.

As the training data size increases, fine-tuning on harder

---

[1]The boundary refers to the difficulty where SFT performance matches that of the base model on a certain data size.

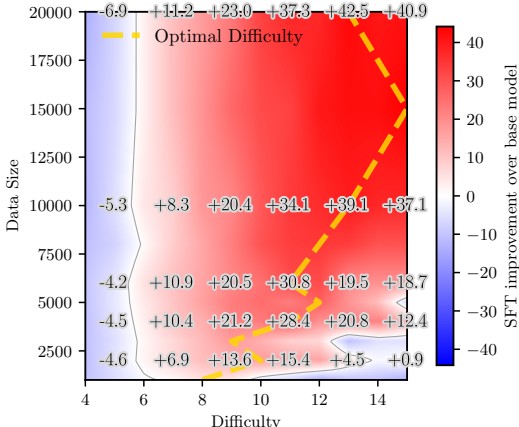
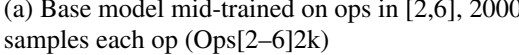

(a) Base model mid-trained on ops in [2,6], 2000 samples each op (Ops[2–6]2k)

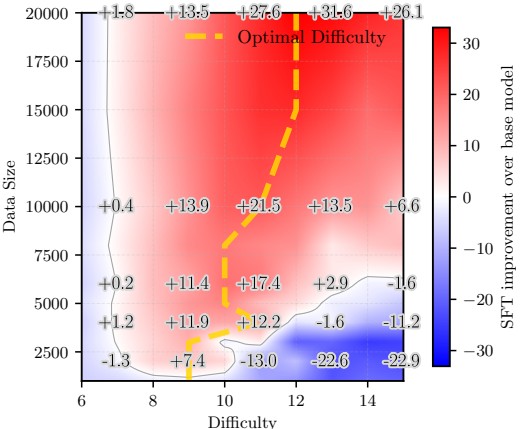

(b) Base model mid-trained on ops in [2,8], 2000 samples each op (Ops[2–8]2k)

Figure 5. Performance gains over different base models on synthetic iGSM data as a function of data difficulty and data size. Data difficulty is measured by the number of operations (ops), and data size corresponds to the number of training samples.

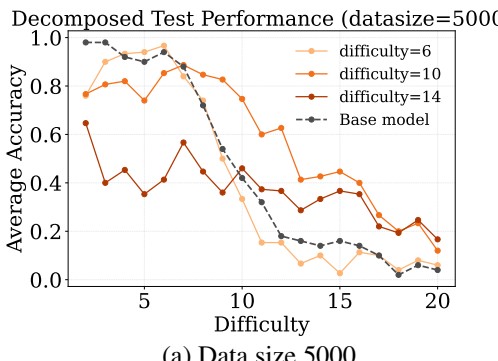

(a) Data size 5000

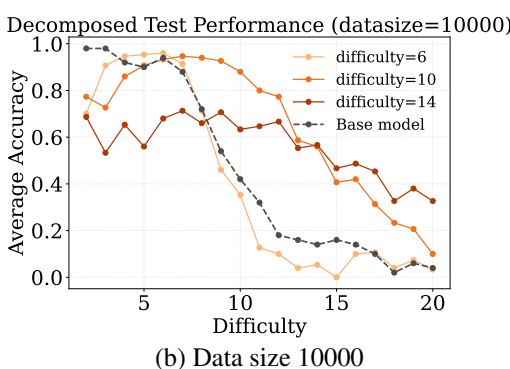

(b) Data size 10000

Figure 6. Decomposed test results for SFT experiments on the base model Ops[2–8]2k under data sizes of 5k and 10k. Each curve represents an SFT model trained at a fixed difficulty (6, 10, or 14), evaluated across test difficulties from 2 to 20.

examples begins to improve performance on harder test instances, whereas fine-tuning on easy data yields limited gains. These observations motivate a two-gap perspective on SFT. We formalize it in the following and use it to explain the behaviors observed above.

Specifically, let $\mathcal{D}_{\text{train}}$ and $\mathcal{D}_{\text{test}}$ denote the post-training SFT data distribution and the downstream test distribution. The downstream performance of a fine-tuned model can be roughly viewed as the combination of two components, as illustrated in Figure 7.

- **In-distribution generalization gap**[2], which quantifies how well the fine-tuned model generalizes on $\mathcal{D}_{\text{train}}$ beyond the finite SFT data samples.

---

[2]Since extrapolation is a form of out-of-distribution generalization, throughout this paper we use "generalization gap" to refer specifically to the in-distribution (IID) generalization gap on $\mathcal{D}_{\text{train}}$, unless otherwise specified.

- **Extrapolation gap**, which captures the mismatch between $\mathcal{D}_{\text{train}}$ and $\mathcal{D}_{\text{test}}$, i.e., how far one must "extrapolate" from the post-training distribution to perform well on the test distribution.

Note that in all experiments the training loss has already converged, suggesting that the degradation observed on very hard data under limited data budgets is not caused by simple optimization non-convergence. Instead, it mainly comes from poor in-distribution generalization, which induces forgetting of previously learned capabilities.

### 4.2.2. EVIDENCE FROM LEARNING THEORY

We further provide a formal setup in statistical learning that models this decomposition, which in turn provides insight into the observations in Section 4.1.

**Data Distributions and Risks.** Let $\mathcal{D}_{\text{train}}$, and $\mathcal{D}_{\text{test}}$ denote the post-training and test distributions over $\mathcal{Z} :=$

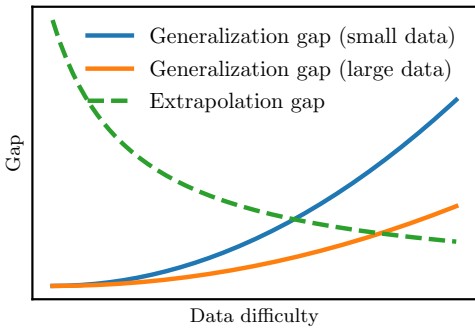
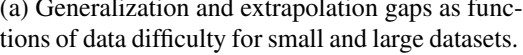

(a) Generalization and extrapolation gaps as functions of data difficulty for small and large datasets.

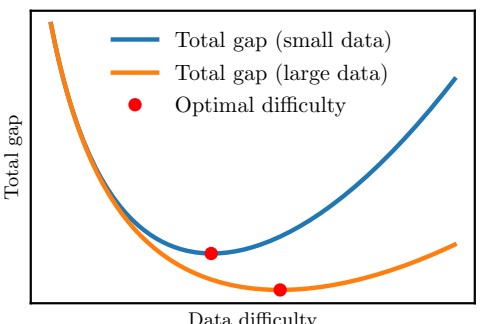

(b) Total performance gap (sum of generalization and extrapolation gaps).

Figure 7. Illustration of the two-gap decomposition in SFT. The generalization gap rises with difficulty, while the extrapolation gap falls. This trade-off gives rise to an interior optimal difficulty. Larger datasets reduce the generalization gap and shift the optimal difficulty toward harder data.

$\mathcal{X} \times \mathcal{Y}$, respectively. We view a language model as an autoregressive conditional family $\{p_\theta(\cdot \mid x)\}_{\theta \in \Theta}$.

For an example $z = (x, y)$ with target sequence $y = (y_1, \ldots, y_T)$, we use the length-normalized token NLL

$$\ell(\theta; z) := -\frac{1}{T} \sum_{t=1}^{T} \log p_\theta(y_t \mid x, y_{<t}). \quad (1)$$

For any distribution $\mathcal{D}$ over $\mathcal{Z}$ and sample set $S$ sampled from $\mathcal{Z}$, define the population and empirical risks as

$$R_\mathcal{D}(\theta) := \mathbb{E}_{z \sim \mathcal{D}}[\ell(\theta; z)], \widehat{R}_S(\theta) := \frac{1}{|S|} \sum_{z \in S} \ell(\theta; z). \quad (2)$$

**Parameter Distributions.** Although in experiments the base model is a fixed parameter vector, for analysis it is convenient to model the outcome of pre-training as a random variable $\theta_{\text{pre}}$ whose randomness captures sources such as random initialization and stochastic optimization. We denote the distribution of $\theta_{\text{pre}}$ by $\pi_{\text{pre}}$. Similarly, we take into consideration the randomness in post-training algorithms. Given a set of post-training samples $S \sim \mathcal{D}_{\text{train}}^n$, we denote the parameter yielded from a post-training algorithm by a random variable $\theta_{\text{train}}$; We denote the distribution of $\theta_{\text{train}}$ by $\pi_{\text{train}}$.

Now we are ready to state the formal characterization of the two-gap decomposition. See Appendix D for a quick proof.

**Proposition 4.1** (Two-gap upper bound). *Fix $\delta \in (0, 1)$ and $n \in \mathbb{N}$. Let $\pi_{\text{pre}}$ be a distribution over $\Theta$ that is independent of the sample set $S$. Assume the loss is uniformly bounded: there exists $C > 0$ such that $\ell(\theta; z) \in [0, C]$ for all $\theta \in \Theta$ and $z \in \mathcal{Z}$. Draw $S = \{z_i\}_{i=1}^{n} \overset{\text{i.i.d.}}{\sim} \mathcal{D}_{\text{train}}^n$. Then with probability at least $1 - \delta$ over the draw of $S$, the following*

*holds simultaneously for all posteriors $\pi_{\text{train}}$ over $\Theta$ (which may depend on $S$):*

$$\mathbb{E}_{\theta \sim \pi_{\text{train}}}\big[R_{\mathcal{D}_{\text{test}}}(\theta)\big] \leq \mathbb{E}_{\theta \sim \pi_{\text{train}}}\big[\widehat{R}_S(\theta)\big]$$
$$+ C\sqrt{\frac{\text{KL}(\pi_{\text{train}} \| \pi_{\text{pre}}) + \ln\left(\frac{1}{\delta}\right) + \frac{5}{2}\ln(n) + 8}{2n - 1}}$$
$$+ C \cdot \text{TV}\big(\mathcal{D}_{\text{test}}, \mathcal{D}_{\text{train}}\big). \quad (3)$$

By Pinsker's inequality, the last term can also be upper-bounded by $C\sqrt{\text{KL}(\mathcal{D}_{\text{test}} \| \mathcal{D}_{\text{train}})/2}$. We use the TV form as the primitive statement because it remains well-defined even under large distribution shifts, while the KL form is most meaningful when $\mathcal{D}_{\text{test}}$ is a moderate shift of $\mathcal{D}_{\text{train}}$ and is absolutely continuous with respect to it. This KL relaxation is standard in domain adaptation analyses (Nguyen et al., 2021).

Proposition 4.1 suggests that, with high probability $\delta$, the gap between the expected risks at train time and test time can be written in a form of

$$\mathbb{E}_{\theta \sim \pi_{\text{train}}}\big[R_{\mathcal{D}_{\text{test}}}(\theta) - \widehat{R}_S(\theta)\big] \leq G_{\text{gen}} + G_{\text{ext}} + \epsilon,$$

where $G_{\text{gen}} = \mathcal{O}(\sqrt{\text{KL}(\pi_{\text{train}} \| \pi_{\text{pre}})/n})$ upper-bounds the in-distribution generalization gap, $G_{\text{ext}} = \mathcal{O}(\text{TV}(\mathcal{D}_{\text{test}}, \mathcal{D}_{\text{train}}))$ upper-bounds the extrapolation gap, and the residual term $\epsilon = \mathcal{O}(\sqrt{\ln(n/\delta)/n})$. Equivalently, under the KL relaxation above, $G_{\text{ext}} = \mathcal{O}(\sqrt{\text{KL}(\mathcal{D}_{\text{test}} \| \mathcal{D}_{\text{train}})})$. Taking derivatives shows that $\epsilon$ decreases with $n$ as long as $n \geq 3 = \lceil e \rceil$, a threshold that all data sizes in our experiments far exceed. Next, we provide qualitative explanations for all four observations mentioned in Section 4.1.

## 4.3. Explaining Empirical Observations

Proposition 4.1 follows from known PAC-Bayesian generalization bounds but provides a decomposition adapted to the LLM SFT scenario, where the post-training distribution can differ substantially from the downstream test distribution. Importantly, the proposition itself only states that the bound depends on the posterior-prior KL term and the train-test distribution shift; it does not imply that harder data must inherently increase the KL term. Our use of the bound is therefore interpretive: for a fixed pre-trained prior, a harder adaptation task may require the learned posterior to incorporate more task-specific information beyond the base model, which can result in a larger posterior-prior divergence. This view is consistent with the standard PAC-Bayesian interpretation of posterior-prior KL as a complexity term, as well as with information-theoretic perspectives on task complexity, memorized information, and task representations (Achille et al., 2021; Harutyunyan et al., 2020; Achille et al., 2019).

Under this interpretation, the two leading terms can vary in opposite directions as training difficulty increases. The generalization term $G_{\mathrm{gen}} = \mathcal{O}(\sqrt{\mathrm{KL}(\pi_{\mathrm{train}}\|\pi_{\mathrm{pre}})/n})$ may increase when fitting harder data requires a larger task-specific adaptation from the base model. On the other hand, the extrapolation term $G_{\mathrm{ext}} = \mathcal{O}(\mathrm{TV}(\mathcal{D}_{\mathrm{test}}, \mathcal{D}_{\mathrm{train}}))$, or its KL relaxation, tends to decrease when harder training data better cover the downstream test cases. Together, these two terms provide a simple explanation for the observed empirical behaviors, as illustrated in Figure 7, and discussed below.

**Explanation of Observation 1 (Data size effect).** When we fix the difficulty of the training data, $\mathcal{D}_{\mathrm{train}}$ is fixed. Increasing the data size $n$ reduces $G_{\mathrm{gen}}$ and the residual term $\epsilon$, while leaving $G_{\mathrm{ext}}$ essentially unchanged since it depends only on the mismatch between $\mathcal{D}_{\mathrm{train}}$ and $\mathcal{D}_{\mathrm{test}}$. Therefore, the test risk decreases as $n$ grows. Once $n$ is large enough, the generalization-related terms become negligible and the remaining error is dominated by the irreducible extrapolation gap $G_{\mathrm{ext}}$, leading to saturation.

**Explanation of Observation 2 (Non-monotonic difficulty effect).** When $n$ is fixed, varying the difficulty changes the post-training distribution $\mathcal{D}_{\mathrm{train}}$ and therefore the SFT solution distribution $\pi_{\mathrm{train}}$. When post-training data are overly easy, $\mathcal{D}_{\mathrm{train}}$ can be poorly aligned with the downstream test distribution $\mathcal{D}_{\mathrm{test}}$, resulting in a large extrapolation gap $G_{\mathrm{ext}}$. As the difficulty increases in this regime, $\mathcal{D}_{\mathrm{train}}$ becomes closer to the harder components of $\mathcal{D}_{\mathrm{test}}$, so $G_{\mathrm{ext}}$ tends to decrease. However, learning from harder post-training examples may require making larger changes to the base model to fit the training data, which can increase $\mathrm{KL}(\pi_{\mathrm{train}}\|\pi_{\mathrm{pre}})$ and thus enlarge $G_{\mathrm{gen}}$. The tradeoff between a decreasing $G_{\mathrm{ext}}$ and an increasing $G_{\mathrm{gen}}$ yields a unimodal dependence on difficulty with an interior optimum.

**Explanation of Observation 3 (Shifting optimal difficulty).** As the data size $n$ increases, $G_{\mathrm{gen}}$ shrinks due to its $1/\sqrt{n}$ dependence, for any fixed post-training distribution $\mathcal{D}_{\mathrm{train}}$. In contrast, $G_{\mathrm{ext}}$ is controlled by the mismatch between $\mathcal{D}_{\mathrm{train}}$ and $\mathcal{D}_{\mathrm{test}}$ and is insensitive to $n$. Therefore, with larger $n$, the cost of learning from harder post-training distributions (described by $G_{\mathrm{gen}}$) becomes less restrictive, and the optimum shifts towards harder $\mathcal{D}_{\mathrm{train}}$ that better reduce $G_{\mathrm{ext}}$. This explains why the empirically optimal difficulty increases with the available data budget.

**Explanation of Observation 4 (Model-relative difficulty).** Difficulty should be interpreted relative to the base model. A stronger base model corresponds to a more informative prior $\pi_{\mathrm{pre}}$, which means that fitting a given post-training distribution may require a smaller change in the induced posterior $\pi_{\mathrm{train}}$, leading to a smaller $\mathrm{KL}(\pi_{\mathrm{train}}\|\pi_{\mathrm{pre}})$ and hence a smaller $G_{\mathrm{gen}}$. At the same time, the extrapolation term $G_{\mathrm{ext}}$ is defined at the distribution level and does not explicitly depend on model capability. As a result, improving the base model primarily relaxes the generalization-side constraint, allowing the model to benefit from harder post-training data without incurring an excessive generalization penalty. Consequently, the optimal difficulty shifts towards higher difficulty as the base model becomes stronger.

## 5. Beyond Sentence-Level Data Selection: A Case Study on DFT

In addition to selecting data at sentence level, recent studies have also explored token-level mechanisms that implicitly modulate training difficulty based on model confidence. A representative recent example is Dynamic Fine-Tuning (DFT) (Wu et al., 2025), which reweights token-level gradients according to token probabilities in order to mitigate the skewed reward signal inherent in SFT. Concretely, by scaling $\nabla \log p_\theta$ with a factor $sg(p_\theta)$, DFT emphasizes tokens that the model already assigns higher probability, i.e., tokens that are easy from the model's perspective. Although DFT has been reported to produce consistent improvements over standard SFT in reasoning tasks, subsequent studies suggest that such gains are context-dependent and not universally guaranteed (Li et al., 2025).

To reconcile these findings and examine whether DFT can alleviate the overfitting and catastrophic forgetting observed in SFT when training on overly easy or overly difficult data, we evaluate DFT under the same controlled synthetic setup and analyze how its behavior compares to SFT across different data sizes and difficulty levels.

Figure 8 shows that DFT exhibits trends similar to SFT,

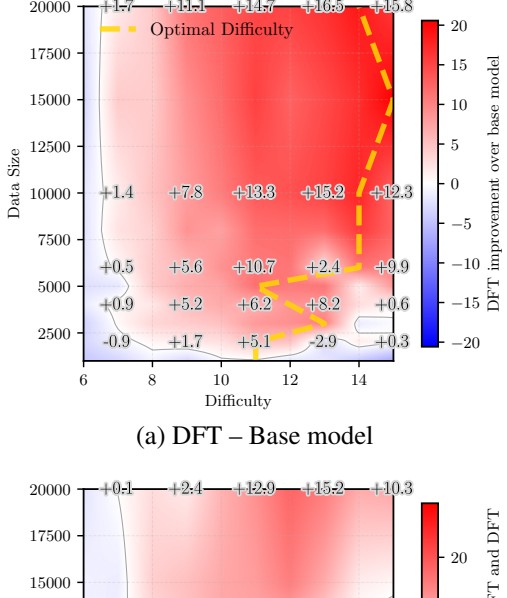

(a) DFT – Base model

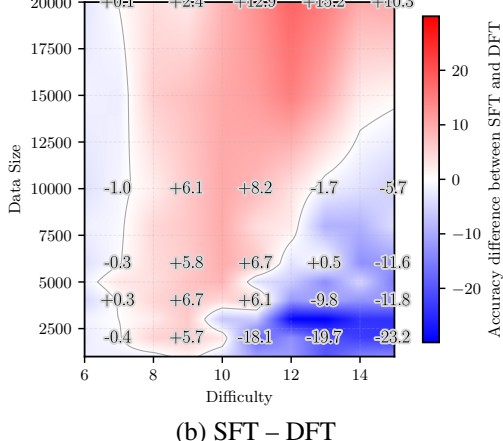

(b) SFT – DFT

Figure 8. DFT performance on synthetic iGSM data (base model Ops[2–8]2k) across various data difficulties and data sizes.

such as performance improving with increasing data size before eventually saturating. However, because DFT favors easy tokens through its token-level reweighting objective, it incurs a smaller generalization gap. As a result, DFT is less sensitive to data difficulty and can outperform SFT when the training data are too difficult relative to the limited data size.

However, due to its objective bias toward high-probability tokens, DFT has limited capacity to effectively fit highly complex data. Consequently, its performance saturates more quickly as data size increases and is eventually surpassed by SFT once sufficient training data are available.

## 6. Conclusion

We study how data difficulty and data size jointly affect supervised fine-tuning. Our experiments show that there exists an optimal data difficulty for SFT, and that this optimal difficulty increases as the available data size grows. This

finding reconciles conflicting results in prior work regarding whether easy or hard data yield better performance. We explain this behavior through the interaction between the IID generalization gap and the extrapolation gap, and support our explanation with controlled synthetic experiments and a PAC-Bayesian theoretical analysis. Overall, our results suggest that effective SFT data selection should adapt difficulty to both data scale and base model capability, rather than relying on fixed filtering heuristics.

Although our work establishes the existence of an optimal data difficulty, identifying it efficiently in practice remains an open problem. In synthetic settings, where the data distribution is low-perplexity and well structured, a simple fitted model can approximate the optimal difficulty. However, real-world data are substantially more complex, making such estimation significantly more challenging. An important direction for future work is to develop efficient methods for estimating the optimal difficulty as a function of the base model, data budget, and target task.

More broadly, our findings suggest a concrete principle for data selection in supervised fine-tuning: rather than uniformly favoring easy or hard data, practitioners should match data difficulty to both model capability and available training scale. From a practical perspective, this suggests that adaptive search strategies over data difficulty may provide an effective way to optimize SFT performance under a fixed data budget, analogous to how regularization and model capacity are selected in classical machine learning. Our framework also offers a useful perspective for understanding why approaches such as data mixture and curriculum learning are often effective, which may help guide future research in these directions.

## Acknowledgements

Jingzhao Zhang is supported by National Key R&D Program of China 2024YFA1015800 and Xiongan AI institute.

## Impact Statement

This paper advances the understanding of data selection in supervised fine-tuning for large language models. We focus on the joint effects of data difficulty and data size on learning and generalization, introducing methodological and theoretical analyses to improve training efficiency and model performance. There are many potential societal consequences of our work, none of which we feel must be specifically highlighted here.

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

# A. Detailed experiment setups

## A.1. Setup for Real-World Experiments.

**Datasets and Difficulty Stratification.**    Our training data is derived from the Chain-of-Thought (CoT) subset of Open-Math (Moshkov et al., 2025), comprising approximately 3.2M mathematical reasoning problems, with step-by-step reasoning solutions. Using ground-truth CoT token length as a proxy for reasoning complexity, we partition this subset into seven distinct non-overlapping difficulty buckets: $[0, 1k), [1k, 4k), [4k, 6k), [6k, 8k), [8k, 12k), [12k, 16k), [16k, 20k)$. From each bucket, we curate a randomized pool of 128k samples and uniformly sample subsets of size $N \in \{1k, 2k, 4k, 8k, 16k, 32k, 64k, 128k\}$ for scaling experiments. This stratified design ensures that comparisons across data sizes are performed under identical difficulty distributions, eliminating confounding effects from content distribution shifts.

**Training and Evaluation Settings.**    We conduct supervised fine-tuning (SFT) experiments using Qwen2.5-Math-1.5B and Qwen2.5-Math-7B (Yang et al., 2024). To support long-context supervision, we increase the RoPE theta to 40,000, and expand the window size to 32,768.

All experiments are implemented using the LLaMA-Factory (Zheng et al., 2024) framework. We use the AdamW optimizer with a base learning rate of $5 \times 10^{-5}$, a constant learning rate schedule with a warmup ratio of 0.03, and a global batch size of 64. All models are trained for a single epoch. To support the extensive Chain-of-Thought trajectories in the hardest difficulty buckets, we set both the maximum training sequence length and maximum generation length to 20,480 tokens, preventing truncation in either phase. All models utilize the default chat template with explicit Chain-of-Thought prompting to elicit step-by-step reasoning.

For evaluation, we employ the vLLM inference engine integrated with the Math-Verify package. Our assessment covers three widely adopted benchmarks spanning diverse difficulty levels: Math500 (Hendrycks et al., 2021), AIME24 (American Institute of Mathematics, 2024), and Minerva Math (Lewkowycz et al., 2022). We report the expected *pass@1* accuracy using nucleus sampling (temperature 0.6, top-$p = 0.95$). To ensure statistical stability, performance is averaged over multiple independent generations per problem ($N = 4$ for Math500, $N = 32$ for AIME 2024, and $N = 8$ for Minerva Math).

## A.2. An overview of iGSM data and iGSM setups

The iGSM dataset (Ye et al., 2024) is a synthetic math dataset designed to emulate the structure and reasoning challenges of GSM8K (Cobbe et al., 2021), while being significantly more scalable and controllable. Each problem in iGSM is generated to capture dependencies among parameters in multi-step arithmetic problems. Specifically, iGSM models three types of dependencies among parameters: direct dependencies, instance dependencies, and implicit dependencies, which require hierarchical reasoning and aggregation over multiple entities.

The data generation process consists of two main steps:

1. **Graph Construction:** Each problem is built from a hierarchical category structure and a dependency graph, where instance parameters are connected by directed edges indicating dependencies. Abstract parameters are inherited from the structure and are not explicitly assigned values.

2. **Problem and Solution Generation:** Problems are described in English sentences corresponding to the dependency graph. Solutions follow a Chain-of-Thought (CoT) format, computing parameters in topological order.

The iGSM dataset allows fine-grained control over problem difficulty by varying two aspects: the number of operations in the solution (denoted as $op$) and the number of edges in the dependency graph. For simplicity, in our work we fix the number of edges according to

$$\#\text{edges} = \left\lfloor op \cdot \frac{4}{3} \right\rfloor + 1,$$

so that difficulty is effectively controlled by $op$. Notice that in the iGSM setup, the problem length grows linearly with the number of operations, which is consistent with our length-based difficulty control discussed in previous sections.

In the iGSM experiments, all models are trained with a batch size of 16 and a cosine learning rate schedule with a warmup ratio of 0.03. During the mid-training stage, we use a learning rate of $5 \times 10^{-5}$ and train for 1 epoch. For the SFT and DFT

---

**iGSM dataset**

**Problem:**
The number of each Rucksack's Label Maker equals 22. The number of each Packable Travel Backpack's Hole Puncher equals 15. The number of each Packable Travel Backpack's Calculator equals 7. The number of each Packable Travel Backpack's Binder Clip equals the difference of each Multi-Day Pack's Stationery and each Rucksack's Stationery. The number of each Multi-Day Pack's Calculator equals 8 times as much as each Packable Travel Backpack's Calculator. The number of each Rucksack's Hole Puncher equals 22 times as much as the sum of each Multi-Day Pack's Calculator and each Packable Travel Backpack's Calculator. How many Hole Puncher does Rucksack have?

**Answer:** 6

**Solution:**
Define Packable Travel Backpack's Calculator as L; so L = 7. Define Multi-Day Pack's Calculator as Z; so Z = 8 * L = 8 * 7 = 10. Define Rucksack's Hole Puncher as g; Q = Z + L = 10 + 7 = 17; so g = 22 * Q = 22 * 17 = 6.

---

Figure 9. An example from the iGSM dataset

experiments, the models are trained for 3 epochs. Specifically, SFT experiments use a learning rate of $5 \times 10^{-5}$, while DFT experiments use a learning rate of $1 \times 10^{-5}$. The test data, midtrain data, and training data are sampled independently using different random seeds. All reported performance metrics are averaged over three random seeds.

## B. Beyond CoT Length: Loss and Pass rate as Difficulty Metric

Although data difficulty is an intuitive and widely used characteristic of data, prior work has not provided an explicit definition of this concept, and no universal difficulty metric exists. To ensure consistency across metrics, we consider three intuitive and commonly adopted measures: Chain-of-Thought (CoT) length, the negative log-likelihood (NLL) of the base model on the CoT solution, and the pass rate of a reference model on the problem. In the main experiments, we primarily use CoT length due to its simplicity and widespread adoption in previous work (Cheng et al., 2025; Yu et al., 2025). Here, we additionally present results using the other two metrics.

For the NLL-based experiment, we compute the per-sample NLL on the OpenMath dataset using a frozen Qwen2.5-Math-1.5B model, and partition the data into seven disjoint intervals: $[0.4, 0.6), [0.6, 0.8), [0.8, 1.0), [1.0, 1.2), [1.2, 1.4), [1.4, 1.8)$ and $[1.8, 2.2)$. For the pass rate-based experiment, we use the pass rate annotations provided by the OpenMath dataset, which are obtained from 32 sampled generations of Qwen2.5-Math-72B-Instruct. We define the corresponding failure rate (1-pass rate), and partition the data into four disjoint intervals: $[0.00, 0.25), [0.25, 0.50), [0.50, 0.75),$ and $[0.75, 1.00]$. Then, following Section 3, we replicate the same scaling protocol: for each difficulty bucket, we construct a randomized pool of 128k examples and sample nested subsets of increasing sizes ($N \in \{1k, \ldots, 128k\}$). All training hyperparameters are kept identical to the primary experimental setup.

**Results.** Figure 10 shows performance as a function of data size across difficulty intervals defined by NLL and failure rate. The results closely mirror those obtained using length-based difficulty: performance consistently scales with data volume, while the scaling behavior is strongly influenced by data difficulty.

1. **Existence of an interior optimum.** Consistent with the observations in the main text, downstream performance does not improve monotonically as difficulty decreases. For a fixed data budget, accuracy typically peaks at an intermediate difficulty range, indicating that neither the easiest examples (lowest loss / highest pass rate) nor the hardest examples (highest loss / lowest pass rate) are optimal for supervised fine-tuning.

2. **Data-dependent shift of optimal difficulty.** We observe a clear shift in the optimal difficulty interval as the training dataset size increases. In the low-data regime, performance is maximized by relatively easy examples (e.g., loss interval $[0.6, 0.8)$ or failure-rate interval $[0.00, 0.25)$), where learning is primarily limited by insufficient effective supervision. As the data budget increases, the optimal region shifts toward more difficult examples (e.g., loss interval $[1.2, 1.4)$ or failure-rate interval $[0.25, 0.50)$), since sufficient data allow the model to better control the generalization gap and benefit from the richer learning signals provided by harder examples.

This consistency indicates that **the trade-off between data quantity and difficulty is intrinsic to SFT, not specific to the choice of difficulty metric**.

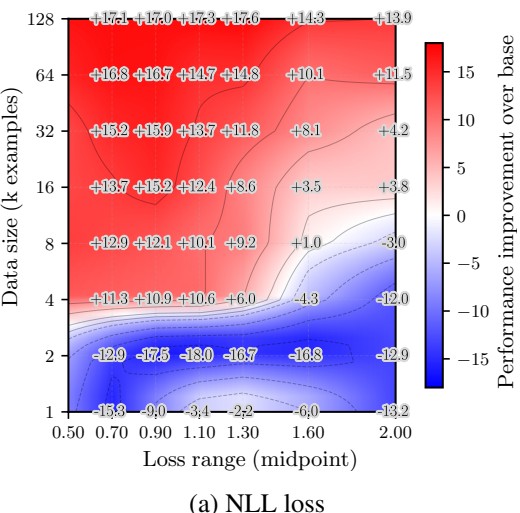

(a) NLL loss

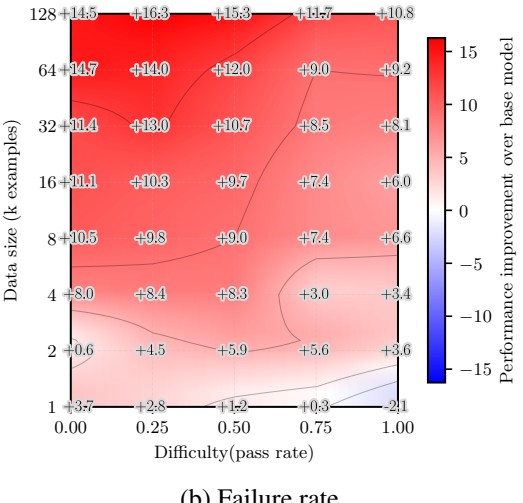

(b) Failure rate

Figure 10. Performance gain over base model as a function of data size and difficulty, trained on the OpenMath dataset, using NLL loss and failure rate as difficulty metrics. Performance is evaluated by the averaged accuracy (%) on Math500, AIME24, and Minerva Math.

## C. Extension to Llama Models and Science Tasks

To evaluate the generality of our findings, we further extend our experiments along two dimensions: model family and task domain. Specifically, we conduct additional experiments using the Llama-3.2-3B model (Grattafiori et al., 2024) and on science reasoning tasks using the OpenScience dataset (NVIDIA, 2025).

For the model extension experiment, we train Llama-3.2-3B on the OpenMath dataset and evaluate performance using the average accuracy over Math500, AIME24, and Minerva Math. For the domain extension experiment, we train Qwen2.5-Math-1.5B on the OpenScience dataset and evaluate performance on MMLU and GPQA. In both cases, data difficulty is measured using CoT length. We follow the same scaling protocol as in the main experiments: from each difficulty bucket, we construct a randomized pool of 128k examples and sample nested subsets of increasing sizes, while keeping all training hyperparameters unchanged. For science evaluation, we report averaged accuracy over multiple independent generations per problem ($N = 2$ for MMLU and $N = 4$ for GPQA).

As shown in Figure 11, both experiments exhibit trends highly consistent with those observed in the main experiments on Qwen models and math reasoning tasks. In particular, we continue to observe a non-monotonic relationship between difficulty and downstream performance, as well as a shift of the optimal difficulty region with increasing data size. These results suggest that our findings are robust across both model families and application domains, rather than being artifacts specific to a particular model or domain.

## D. A Proof of Proposition 4.1

To prove Proposition 4.1, we divide the discrepancy $\mathbb{E}_{\theta \sim \pi_{\text{train}}}\big[R_{\mathcal{D}_{\text{test}}}(\theta)\big] - \mathbb{E}_{\theta \sim \pi_{\text{train}}}\big[\widehat{R}_S(\theta)\big]$ into two gaps: (i) in-distribution generalization under $\mathcal{D}_{\text{train}}$ and (ii) extrapolation from $\mathcal{D}_{\text{train}}$ to $\mathcal{D}_{\text{test}}$, which can be upper-bounded respectively by Lemma D.1 and Lemma D.2. We state the extrapolation term first in total variation distance, which is the more primitive form. The full KL version of the two-gap bound follows as a corollary via Pinsker's inequality, and is most meaningful when $\mathcal{D}_{\text{test}}$ is absolutely continuous with respect to $\mathcal{D}_{\text{train}}$.

**Lemma D.1** (PAC-Bayes in-distribution bound; Theorem 3.1 in Alquier (2024)). *Under the assumptions of Proposition 4.1, fix any $\delta \in (0, 1)$. With probability at least $1 - \delta$ over the draw of S, the following holds simultaneously for all posteriors $\pi_{\text{train}}$ over $\Theta$:*

$$\mathbb{E}_{\theta \sim \pi_{\text{train}}}\big[R_{\mathcal{D}_{\text{train}}}(\theta)\big] \ \leq \ \mathbb{E}_{\theta \sim \pi_{\text{train}}}\big[\widehat{R}_S(\theta)\big] + C\sqrt{\frac{\text{KL}(\pi_{\text{train}}\|\pi_{\text{pre}}) + \ln\big(\frac{1}{\delta}\big) + \frac{5}{2}\ln(n) + 8}{2n - 1}}.$$

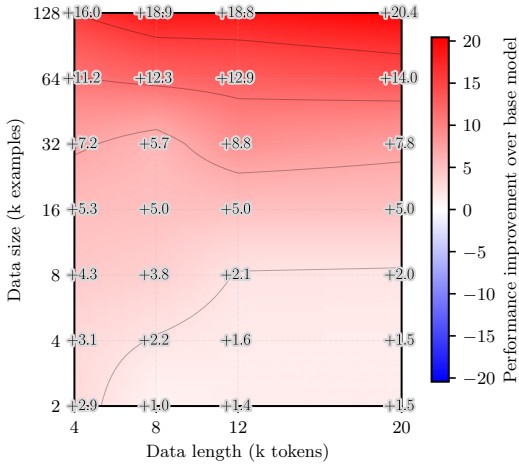
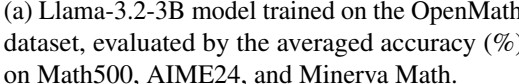

(a) Llama-3.2-3B model trained on the OpenMath dataset, evaluated by the averaged accuracy (%) on Math500, AIME24, and Minerva Math.

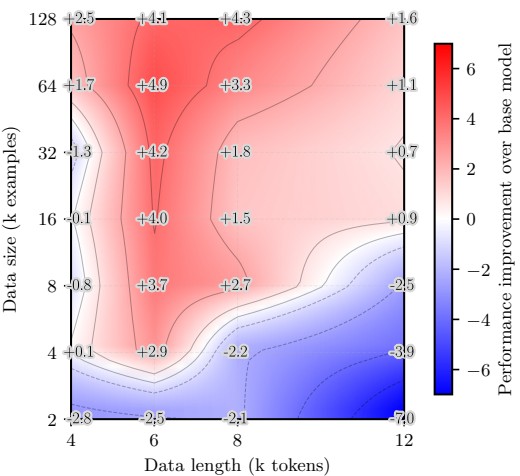

(b) Qwen2.5-Math-1.5B model trained on the OpenScience dataset, evaluated by the averaged accuracy (%) on MMLU and GPQA.

Figure 11. Extension experiments on Llama models and science reasoning tasks. Data difficulty is measured using CoT length.

---

**Remark (The Pac-Bayes Bound).** This bound was originally studied in McAllester (1998) on 0-1 loss functions, and was later generalized to all loss functions taking values in $[0, 1]$ in Maurer (2004). Here, we use the form restated in Alquier (2024). Although the original bound was stated for $C = 1$, applying the original bound to a rescaled version of the loss functions yields the result for arbitrary $C$.

---

Next, we move on to the extrapolation control from $\mathcal{D}_{\text{train}}$ to $\mathcal{D}_{\text{test}}$.

**Lemma D.2.** *Under the assumptions of Proposition 4.1, for any fixed $\theta \in \Theta$,*

$$\left| R_{\mathcal{D}_{\text{test}}}(\theta) - R_{\mathcal{D}_{\text{train}}}(\theta) \right| \leq C \cdot \text{TV}\big(\mathcal{D}_{\text{test}}, \mathcal{D}_{\text{train}}\big). \tag{4}$$

*Proof.* Fix $\theta \in \Theta$ and write $f(z) := \ell(\theta; z) \in [0, C]$. By the variational characterization of total variation distance,

$$\left| \mathbb{E}_{z \sim \mathcal{D}_{\text{test}}}[f(z)] - \mathbb{E}_{z \sim \mathcal{D}_{\text{train}}}[f(z)] \right| \leq C \cdot \text{TV}\big(\mathcal{D}_{\text{test}}, \mathcal{D}_{\text{train}}\big), \tag{5}$$

where $\text{TV}(P, Q) := \sup_{A \subseteq \mathcal{Z}} |P(A) - Q(A)|$. □

Finally, combining Lemma D.1 and Lemma D.2 gives exactly Proposition 4.1. Corollary D.3 gives the KL-form version of the same two-gap upper bound.

**Corollary D.3** (Two-gap KL upper bound). *Fix $\delta \in (0, 1)$ and $n \in \mathbb{N}$. Let $\pi_{\text{pre}}$ be a distribution over $\Theta$ that is independent of the sample set $S$. Assume the loss is uniformly bounded: there exists $C > 0$ such that $\ell(\theta; z) \in [0, C]$ for all $\theta \in \Theta$ and $z \in \mathcal{Z}$. Draw $S = \{z_i\}_{i=1}^{n} \overset{\text{i.i.d.}}{\sim} \mathcal{D}_{\text{train}}^{n}$. If $\text{KL}(\mathcal{D}_{\text{test}} \| \mathcal{D}_{\text{train}})$ is finite, then with probability at least $1 - \delta$ over the draw of $S$, the following holds simultaneously for all posteriors $\pi_{\text{train}}$ over $\Theta$ (which may depend on $S$):*

$$\mathbb{E}_{\theta \sim \pi_{\text{train}}}\big[ R_{\mathcal{D}_{\text{test}}}(\theta) \big] \leq \mathbb{E}_{\theta \sim \pi_{\text{train}}}\big[ \widehat{R}_S(\theta) \big]$$
$$+ C\sqrt{\frac{\text{KL}(\pi_{\text{train}} \| \pi_{\text{pre}}) + \ln\left(\frac{1}{\delta}\right) + \frac{5}{2}\ln(n) + 8}{2n - 1}}$$
$$+ C\sqrt{\frac{1}{2}\text{KL}\big(\mathcal{D}_{\text{test}} \| \mathcal{D}_{\text{train}}\big)}. \tag{6}$$

*Proof.* This follows by combining Lemma D.1 with Lemma D.2 and then applying Pinsker's inequality to the TV term. □

**Discussion of the TV and KL forms.** The TV bound in Proposition 4.1 is the more primitive statement: it follows directly from the variational characterization of total variation distance and only requires bounded losses. It is also well-defined under large distribution shifts, including cases where the support of $\mathcal{D}_{\text{test}}$ is not fully covered by $\mathcal{D}_{\text{train}}$. By contrast, the KL form in Corollary D.3 is obtained by relaxing TV with Pinsker's inequality. This form is useful for exposition because it expresses both terms in familiar KL quantities and aligns with common domain-adaptation analyses (Nguyen et al., 2021). However, it should be interpreted as a corollary rather than the fundamental statement: when $\mathcal{D}_{\text{test}}$ is far from $\mathcal{D}_{\text{train}}$, or is not absolutely continuous with respect to it, the KL term may become loose or even vacuous. Thus, we use the TV bound as the main theorem-level control and the KL bound as an interpretable relaxation for moderate distribution shifts.

