# OpenReview forum: "Data Difficulty and the Generalization–Extrapolation Tradeoff in LLM Fine-Tuning"
_ICML.cc/2026/Conference — ICML 2026 regular_

### Official Review · Reviewer_5o7C · 2026-03-12

**Soundness:** 2
**Presentation:** 3
**Significance:** 2
**Originality:** 3
**Overall Recommendation:** 4
**Confidence:** 4

**Summary:**

This paper studies how data difficulty and dataset size jointly affect SFT of LLMs. The central finding is that there exists an optimal data difficulty for SFT that shifts toward harder examples as the data budget grows. The authors first demonstrate this empirically on real math datasets, including OpenMath, OpenR1-Math, with Qwen2.5-Math models, then reproduce the pattern in controlled synthetic experiments using iGSM grade-school math data. They propose a two-gap explanation: the in-distribution generalization gap (increases with difficulty) trades off against the extrapolation gap (decreases with difficulty), and more data shrinks the generalization gap, shifting the optimum toward harder data. A PAC-Bayesian bound formalizes this decomposition. A brief case study on Dynamic Fine-Tuning (DFT) shows similar trends but with earlier saturation due to its bias toward easy tokens.

**Compliance With Llm Reviewing Policy:**

Affirmed.

**Final Justification:**

Based on our conversation with the authors and my reviews, I'd recommend a weak accept.

**Key Questions For Authors:**

1. The theoretical analysis (Proposition 4.1) combines a standard PAC-Bayes bound with Pinsker's inequality: both are farily well-known results. The combination is sensible but somewhat loose. For example, the extrapolation term uses KL(D_test || D_train), which is infinite whenever the test distribution has support outside the training distribution (can be pretty common empirically). How should one interpret the bound in the realistic SFT setting where hard test problems may lie entirely outside the training support?

2. The difficulty metric used throughout (CoT length) conflates reasoning complexity with verbosity. Appendix B shows loss-based difficulty yields similar trends, which is reassuring, but have you considered cases where the two metrics diverge substantially (e.g., long but formulaic solutions vs. short but tricky problems)? How sensitive are the main conclusions to the choice of difficulty proxy?

3. Main experiments are on mathematical reasoning. I can see that being the most easy in terms of establishing difficulty. How should difficulty be intepreted in other settings? Do authors have any evidence, even preliminary, that the generalization–extrapolation tradeoff structure holds for other SFT tasks (e.g., instruction following, code generation, or open-ended QA)?

4. As noted in my earlier reviews. How empirically useful is the claim that "there exists an optimal difficulty level for SFT"? The paper poses the practical challenge of selecting SFT data, but it is hard to see the value beyond knowing this optimum exists. Can we extrapolate or predict the optimal ratio?

**Limitations:**

Some of the limitations are discussed as what i’ve noted in the weakness.

On the whole, the main limitation is that paper's experimental scope is narrow: all findings are derived from a single model family (Qwen2.5-Math) on mathematical reasoning tasks only, making it unclear how well the conclusions transfer. The theoretical bound, while clean, is a direct combination of known results and can be questionable when test and train supports diverge. Finally, the work identifies the existence of an optimal difficulty but offers no practical method to estimate it, limiting its empirical usefulness.

**Strengths And Weaknesses:**

**Strengths**

- The claims are quite interesting and useful for post-training data selection (specifically SFT): optimal data difficulty is a function of training size.
- Controlled synthetic experiments give the authors a fairly controlled way to manipulate problem difficulty (however, mainly in grade-school math, which already limits the range of difficulty).
- The interaction between generalization and extrapolation gap is neat and useful.

**Weaknesses**

- How empirically useful is the claim that "there exists an optimal difficulty level for SFT"? The paper poses the practical challenge of selecting SFT data, but it is hard to see the value beyond knowing this optimum exists. Can we extrapolate or predict the optimal ratio? The relationship seems highly unpredictable judging by the optimal difficulty curves (Figure 5). This is also partly acknowledged by the authors, who note that identifying the optimal difficulty in practice remains an open problem.

- The core claim of the paper and the scale of the experiments are somewhat mis-aligned. The core claim is that for a fixed data budget, there exists an optimal data difficulty for SFT. However, model choice seems to play an equally critical role, if not a larger role, in data difficulty. Regarding data difficulty, one should agree that the optimal data difficulty is not only a function of dataset size but also dependent on pre-trained model choice (for example, how capable the model is). The authors do discuss this as "Observation 4" in the iGSM section and compare 1.5B vs 7B in Figure 3, but this discussion is largely qualitative and limited to two sizes within the same model family. Given the fact that the experiments are done exclusively on the Qwen2.5-Math series of models, one does wonder how the findings generalize. This is especially true given recent literature discussing how model choice is critical to data choice (e.g., Spurious Rewards: Rethinking Training Signals in RLVR).

---

> ### Author Rebuttal · Authors · 2026-03-31
>
> Thank you for your thoughtful and detailed feedback. Your comments and suggestions are very helpful for refining our work.
>
> 1. **Using pass rate as difficulty metric**. Based on Q2 regarding difficulty metric, we added experiments using the pass rate as metric. The results are shown below, which are consistent with our current results.
>
> https://files.catbox.moe/bm3867.pdf
>
> **Rebuttal Figure 1**: 2D performance improvement over base model Qwen2.5-Math-1.5B, trained on OpenMath, evaluated by average accuracy on Math500, AIME24, and Minerva Math. Difficulty is measured by (1.0 - pass rate).
>
> 2. **Extension to science tasks and another model family**. By the suggestion on generalization to different tasks in Q3 and generalization to different model family in W2, we added experiments on science task and Llama model respectively. The results are shown below. Model performance exhibits similar trends as main experiments.
>
> https://files.catbox.moe/uba5m5.pdf
>
> **Rebuttal Figure 2**: 2D improvement over base model Qwen2.5-Math-1.5B, trained on OpenScience, evaluated by average accuracy on MMLU and GPQA. Difficulty is measured by CoT length.
>
> https://files.catbox.moe/svb93a.pdf
>
> **Rebuttal Figure 3**: 2D improvement over base model Llama-3.2-3B, trained on OpenMath, evaluated by average accuracy on Math500, AIME24, and Minerva Math. Difficulty is measured by CoT length.
>
> Below, we discuss each comments and questions in more details.
>
> ## Response to Q2 (Sensitivity to Difficulty Measure)
> >Q2. ... How sensitive are the main conclusions to the choice of difficulty proxy?
> >
> Thank you for raising this point. We define difficulty in a common-sense way. It measures  how difficult the **question** is for the model and how easy it is for the model to learn the **solution**.  Since no universal metric exists in prior works, we adopt three widely used proxies: **CoT length**, **NLL loss** and **pass rate**.  All three results exhibit similar trends(Section 3, Figure 3,4; Appendix B, Figure 10; Rebuttal Figure 1).  The consistency of our findings across metrics ensures that **the observed effects stem from difficulty itself rather than from any particular proxy**.
>
> We will add experiments and discussion for metric selection explicitly to the main text.
>
> ## Response to Q3 (Generalization Across Tasks)
> >Q3.  ... Do authors have any evidence that ... holds for other SFT tasks?
>
> To check whether our findings generalize across tasks, we conduct another 2D experiments on the science task. The results are shown in Rebuttal Figure 2 which align well with our current findings. We will add these experiments into the refined version.
>
> ## Response to W2 (Dependency of Optimal Difficulty on Model)
>
> Thank you for raising this point.  As discussed in Q2, we understand difficulty as a property **relative to a specific model**, so our conclusion holds cleanly without ambiguity in this sense.
>
> Based on your concerns about whether the results generalize to different model families, we conducted 2D experiments on Llama3.2-3B model. The results are shown in Rebuttal Figure 3 which exhibit similar performance, ensuring generalizability.
>
> We will add our experiments on different model families to the main text.
>
> ## Response to Q1 (Loose Bound for the KL Term)
> >Q1. ...The combination is sensible but somewhat loose...where hard test problems may lie entirely outside the training support?
> >
> We agree that this KL form is mainly meaningful when $D_{test}$ is a moderate shift of $D_{train}$.
>
> At the same time, **the more primitive statement is the corresponding total-variation bound $\mathrm{TV}(D_{test},D_{train})$** which is well-defined even under large distribution shift. **We originally presented the result in KL form for exposition**, as it decomposes naturally into KL terms and is standard in prior work [1]. However, we agree that the TV form is the more fundamental statement and will highlight the TV bound explicitly and present the KL form as a corollary.
>
> [1] Nguyen et al. “KL Guided Domain Adaptation.” arXiv preprint arXiv:2106.07780, 2021.
>
> ## Response to Q4 & W1 (Predicting the Optimal Difficulty)
> >W1. How empirically useful is the claim that "there exists an optimal difficulty level for SFT"? ... Can we extrapolate or predict the optimal ratio?
> >
> Thank you for pressing on this important question. The optimal difficulty depends on many factors(e.g. pretraining data) which are difficult to quantify, so  predicting the optimal difficulty is not yet realistic.
>
> Despite of this, we believe **our findings provide a concrete guiding principle for data selection**. One possible application is  to do binary search on the optimal difficulty, **just as our knowledge in overfitting help selecting model regularization and capacity in standard Machine Learning**. Our framework also offers a useful lens for understanding why data mixture and curriculum learning work.
>
> We will discuss this more explicitly in the contribution.

---

> > ### Author Rebuttal · Reviewer_5o7C · 2026-04-01
> >
> > Thank you to the authors for addressing all of my questions. The additional empirical results provided during the rebuttal are encouraging. However, my core concerns remain. The core findings that there exists an optimum data difficulty given data sizes and difficulty tends to scale as data scale, while useful, are not particularly surprising, and the paper lacks the ability to extrapolate beyond the settings studied. For these reasons, my score stays the same.

---

> > > ### Author Response · Authors · 2026-04-04
> > >
> > > Thank you for the thoughtful engagement and for acknowledging the additional experiments. We agree that translating these findings into concrete predictions is an important open problem, and we see leveraging our framework to better understand methods like data mixture and curriculum learning — and ultimately develop more principled SFT pipelines based on difficulty — as a natural and exciting direction for future work. We sincerely appreciate your constructive feedback, which has helped sharpen the scope and presentation of our work.

---

### Official Review · Reviewer_Bk9e · 2026-03-17

**Soundness:** 3
**Presentation:** 3
**Significance:** 2
**Originality:** 3
**Overall Recommendation:** 4
**Confidence:** 4

**Summary:**

This paper systematically investigates how the interplay between data size and data difficulty affects SFT performance in LLMs, and finds that the optimal data difficulty increases as the training data size.

The authors further support their findings from the following aspects:
- Empirical evidence: A series of controlled synthetic experiments demonstrate a consistent pattern in which the optimal difficulty shifts toward harder examples as the data size increases.

- Theoretical analysis: Using the PAC-Bayesian model to provide a principled explanation by decomposing the test error into generalization and extrapolation gaps.

**Compliance With Llm Reviewing Policy:**

Affirmed.

**Final Justification:**

The authors have addressed the concerns and provided additional experimental results, so I increase the overall recommendation score. However, the practical impact and application value of the paper remain limited. Due to the complexity of real-world data (higher noise levels or greater diversity), in my view, it may be difficult to eliminate other factors, which could affect the method’s effectiveness. Therefore, I maintain the original significance score.

**Key Questions For Authors:**

- Can the authors conduct experiments on additional domains, such as code or scientific problem-solving, to test whether the findings generalize beyond the current datasets?
- In Figure 3, why is performance evaluated using the average accuracy across Math500, AIME24, and MinervaMath rather than reporting results separately for each dataset? Will the averaged results lead to inconsistent trends where one dataset shows improvement while another shows a decrease one?
- Can the authors clarify or experimentally isolate the effects of other factors, such as data diversity, entropy, or noise, to strengthen the validity of the findings? For example, comparing training datasets with similar noise or entropy levels but differing in size or difficulty to check whether the optimal points trend remains consistent.

**Limitations:**

See the Weaknesses 3.

**Strengths And Weaknesses:**

Strengths:
- The paper presents a novel finding that there exists an optimal trade-off between data size and data difficulty for SFT performance in LLMs, and that the optimal boundary shifts toward higher difficulty as the data size increases.
- The authors validate their four key observations through both empirical evidence and a formal PAC-Bayesian theoretical analysis. The consistency between these two perspectives strengthens the credibility of their findings.
- The paper is well-organized and easy to follow, with clear figures and sufficiently detailed theoretical derivations.

Weaknesses:
- Limited factor isolation: While the paper emphasizes the role of data size and difficulty, it does not sufficiently control other potentially confounding factors (e.g., data diversity, noise). This raises concerns about whether the reported trends can be cleanly attributed to the proposed variables.
- Limited generalizability: The experiments are primarily conducted on math-related datasets. It remains unclear whether the observed trends generalize to other domains, which may limit the broader applicability.
- Practical limitations: Although the authors claim that their findings pose challenges for real-world data, the paper does not compare with existing data selection methods in the experiments, which are not based on real-world datasets. Without such a comparison, it remains unclear whether the findings offer any practical advantage over alternative methods.

---

> ### Author Rebuttal · Authors · 2026-03-31
>
> We sincerely thank the reviewer for the thoughtful feedback and for recognizing the novelty of our core findings, as well as the consistency between empirical and theoretical results.
> Inspired by your comments, we add new experiments and provide detailed clarifications below.
>
> - **Extension to science tasks.** Additional experiments on **OpenScienceReasoning-2** demonstrate consistent results, which broaden the scope of our work.
>
> https://files.catbox.moe/uba5m5.pdf
>
> **Rebuttal Figure 1**: 2D improvement over base model Qwen2.5-Math-1.5B, evaluated by average accuracy on MMLU and GPQA.
>
> - **Decomposed benchmarks.** We report per-benchmark results for Figure 3(a); the same trend holds across all benchmarks.
>
> **Rebuttal Figure 2(a)**: AIME24 https://files.catbox.moe/n46fcx.pdf
>
> **Rebuttal Figure 2(b)**: Math500 https://files.catbox.moe/xkr0d2.pdf
>
> **Rebuttal Figure 2(c)**: Minerva-Math https://files.catbox.moe/ggwyim.pdf
>
> ## Response to W2 & Q1 (Generalizability)
>
> > W2/Q1. Limited generalizability: ...remains unclear whether the observed trends generalize to other domains...
>
> Thanks for raising this insightful question. To test broader applicability, we conducted 2D experiments on **OpenScienceReasoning-2** following the identical protocol as our real-world math experiments. The results shown in Rebuttal Figure 1 exhibit the same patterns as our main findings. This clearly demonstrates that **the data size-difficulty tradeoff holds true in a broader, multi-domain setting.**
>
> ## Response to W3 (Comparison with Existing Methods)
>
> > W3. Practical limitations: ...does not compare with existing data selection methods...
>
> The practical concern regarding empirical comparisons is well understood. We would like to clarify that our work is orthogonal, and can be applied to existing data selection procedures.
>
> **Orthogonal Objectives:** Existing methods (e.g., LESS, LIMO, s1) typically select data based on static criteria. In contrast, our core contribution establishes that **the optimal difficulty level is non-stationary and increases systematically with data scale.** Because static filtering inherently cannot capture this dynamic shift (Figure 2), direct empirical comparisons conflate fundamentally different research objectives.
>
> **Practical Complementarity:** Rather than competing with prior methods, these findings provide principled guidance to enhance them. Specifically, data selection must be **adaptive to scale**:
>
> - **Limited-data regimes:** Prioritizing overly difficult examples increases extrapolation error and degrades performance.
> - **Large-data regimes:** Incorporating harder examples becomes strictly beneficial, allowing the model to exploit richer supervision signals.
>
> This suggests a simple practical strategy: **progressively expand the difficulty frontier as the dataset grows**, instead of fixing it *a priori*.
>
> ## Response to W1 & Q3 (Factor Isolation)
>
> > W1/Q3. Limited factor isolation: ...does not sufficiently control other potentially confounding factors (e.g., diversity, noise)...
>
> We agree that data diversity and noise are critical factors in SFT performance. Our two-level framework strictly isolates data size and difficulty from these confounding variables:
>
> **Distributional Control (Real-World):** In the real-world experiments, we enforce **identical data provenance**. All problems originate from the same source pool and CoT solutions were generated using the identical models. This ensures that topic distribution and generation quality do not systematically vary across buckets. While residual correlations cannot be entirely eliminated in naturalistic data, **this is precisely why the fully controlled synthetic setting is introduced**.
>
> **Strict Factor Isolation (Synthetic):** To perfectly isolate difficulty, the iGSM setup (Section 4) is utilized. As referenced in PhysicsLM, diversity across instances is consistently minimized to eliminate noise from distributional mismatch. Here, problem structure, linguistic diversity, and noise levels are **held strictly constant** by the rule-based generator and num of operations is the *sole* variable governing difficulty.
>
> Crucially, **all four core observations replicate perfectly in the noise-free iGSM setting**, proving our findings are fundamentally driven by data size and difficulty.
>
> ## Response to Q2 (Averaging Across Benchmarks)
>
> > Q2. ...why is performance evaluated using the average accuracy...Will the averaged results lead to inconsistent trends...?
>
> Thank you for pointing this out. The aggregated metric is utilized strictly for variance reduction as benchmarks like AIME24 inherently exhibit high variance. To address the concern of masked opposing trends, isolated performance for all three benchmarks is provided in Rebuttal Figure 2.
>
> The results show that averaging process does not mask any opposing trends; rather, **all individual benchmarks consistently demonstrate the identical core phenomena** regarding the data size-difficulty tradeoff.

---

> > ### Author Rebuttal · Reviewer_Bk9e · 2026-04-02
> >
> > Thanks to the authors for addressing the questions and concerns with additional experiments and clarifications. Hence, I have increased my overall recommendation score. However, in my view, the practical impact and application value of the paper remain limited (also noted in the Impact Statement), due to the complexity of real-world data (e.g., higher noise levels and greater diversity). Therefore, the significance score of this paper remains unchanged.

---

> > > ### Author Response · Authors · 2026-04-04
> > >
> > > Thank you for raising your score and for the kind acknowledgement. We agree that improving practical applicability is important, and plan to explore this in future work—for example, by developing adaptive difficulty scheduling methods to better optimize SFT performance. We also appreciate your constructive feedback in helping strengthen the paper.

---

### Official Review · Reviewer_tooc · 2026-03-19

**Soundness:** 3
**Presentation:** 3
**Significance:** 3
**Originality:** 3
**Overall Recommendation:** 5
**Confidence:** 3

**Summary:**

The paper studies the relation between data difficulty and dataset size in SFT training of LLMs. The paper shows that the optimal data difficulty is dependent on the total available data budget and the capability of the base model. The paper includes evaluations on math and reasoning datasets and a controlled synthetic experiments using the iGSM dataset. Results show that optimal training difficulty shifts toward harder examples as the data size increases. Finally, the paper proposes a theoretical PAC-Bayesian framework, decomposing the SFT error into two terms: an in-distribution generalization gap (which increases with hard data) and an extrapolation gap (which decreases with hard data).

**Compliance With Llm Reviewing Policy:**

Affirmed.

**Final Justification:**

The responses mostly address my comments.

**Key Questions For Authors:**

- Can you provide empirical measurements of the KL divergence between the base model and the fine-tuned models across different difficulty buckets?
- Would it be feasible to conduct a smaller-scale 2D experiment where the difficulty score is pass-rate rather than CoT length or NLL?
- In Figure 6, you show that SFT on very difficult data causes performance to drop when data is limited. To what extent is this caused by catastrophic forgetting vs. failure to converge?

**Limitations:**

Yes.

**Strengths And Weaknesses:**

**Strengths**
- The paper evaluates both, the synthetic iGSM data and real-world data. Additionally, the paper calibrates the Qwen2.5-Math model to the OOD synthetic data by incorporating a mid-training phase.
- The paper addresses conflicting findings in recent literature regarding prioritization of easy vs. hard data by introducing dataset size as a variable.

**Weaknesses**
- The PAC-Bayesian theory relies on the assumption that training on harder data inherently increases the KL divergence term, without providing empirical support for this assumption.
- The paper lacks a discussion on the limitations of using CoT length as a proxy of difficulty, and whether the correlation between CoT and pass rate could be spurious.

---

> ### Author Rebuttal · Authors · 2026-03-31
>
> Thank you for the thoughtful feedback. We appreciate the suggestion to further clarify both the empirical status of the PAC-Bayesian interpretation and the justification of difficulty metric selection. We will revise the paper accordingly.
>
> **Using pass rate and model likelihood as difficulty metric**. Based on W2 and Q2 regarding difficulty metric, we added experiments using the pass rate as metric. The results are shown below, which are consistent with our current results.
>
> https://files.catbox.moe/bm3867.pdf
>
> **Rebuttal Figure 1**: 2D performance improvement over base model Qwen2.5-Math-1.5B, trained on OpenMath, evaluated by average accuracy on Math500, AIME24, and Minerva Math. Difficulty is measured by (1.0 - pass rate).
>
> Details are below.
>
> ## **Response to Q1/W1 (KL term in Theory)**
>
> > Q1. Can you provide empirical measurements of the KL divergence between the base model and the fine-tuned models across different difficulty buckets?
> >
>
> > W1. The PAC-Bayesian theory relies on the assumption that training on harder data inherently increases the KL divergence term, without providing empirical support for this assumption.
> >
>
> We thank the reviewer for raising these important points. We agree that PAC-Bayesian theory does **not** imply that training on harder data must inherently increase . Rather, our intended interpretation is that for a fixed pre-trained prior , a harder *adaptation task* may require the learned posterior to incorporate more task-specific information beyond the base model, which can result in a larger posterior-prior divergence. This is consistent with the standard PAC-Bayesian view that the posterior-prior KL serves as a complexity term relative to a data-independent prior, as well as with information-theoretic perspectives on transfer learning that relate adaptation difficulty to the amount of task-specific information required beyond a pre-trained initialization [1].
>
> We will clarify that PAC-Bayes only shows that the generalization bound depends on the posterior-prior KL term; connecting this term to task difficulty requires additional assumptions and therefore should be viewed as an **interpretation** rather than an theoretical implication.
>
> Regarding empirical measurements, **the challenge is that  is a divergence between distributions over parameters** rather than between data distributions. For modern neural networks, this quantity is generally intractable since we need to repeat the pretraining stage numerous times. This difficulty is well recognized in the literature: related works typically interpret posterior-prior KL as measures of complexity or memorized information[2]. Likewise, prior work on transfer and task representation supports the broader idea that different target tasks can require different amounts of task-specific information[1,3].
>
> We will revise the paper to **make the theoretical claim more precise and avoid overstating**.
>
> [1] Achille et al. "The information complexity of learning tasks, their structure and their distance." arXiv preprint arXiv:1904.03292, 2019.
>
> [2] Harutyunyan et al. "Improving generalization by controlling label-noise information in neural network weights." In Proceedings of the 37th International Conference on Machine Learning, 2020.
>
> [3] Achille et al.. "Task2Vec: Task embedding for meta-learning." In Proceedings of the IEEE/CVF International Conference on Computer Vision, 2019.
>
> ## **Response to Q3 (Failure Mode on Very Hard Data).**
>
> > Q3. In Figure 6, you show that SFT on very difficult data causes performance to drop when data is limited. To what extent is this caused by catastrophic forgetting vs. failure to converge?
> >
>
> The degradation on very hard data under limited data is **driven mainly by forgetting, rather than by optimization non-convergence.** The clearest signal is that performance drops even on easy and medium test slices that the base model already handles well, indicating forgetting. At the same time, the effect is **not purely forgetting** since with limited hard data, there is also a clear in-distribution generalization bottleneck on the hard region. We avoid the wording “failure to converge” as the loss curves have all converged; instead, we will describe it as **limited in-distribution generalization** under scarce hard-data supervision. We will discuss this distinction explicit in the revision.
>
> ## **Response to W2 / Q2 (difficulty metric and robustness).**
>
> To check whether our finding generalize across metrics, we have conducted experiments using **NLL loss** and **pass rate** for difficulty beyond CoT length which exhibit consistent trends(Section 3, Figure 3,4; Appendix B, Figure 10; Rebuttal Figure 1). The consistency among metrics suggests that **our conclusions are not an artifact of a single proxy, but reflect a general relationship between data difficulty and data size**.

---

> > ### Author Rebuttal · Reviewer_tooc · 2026-04-03
> >
> > I thank the authors for the additional experiments and responses to my questions. Most of my comments are adequately addressed. However, I remain unsure on the significance and utility of the proposed BAC-Bayes theory.

---

> > > ### Author Response · Authors · 2026-04-04
> > >
> > > Thank you for your generous score and for the continued engagement with our work.
> > >
> > > Regarding the PAC-Bayesian perspective, we use it as a simple, principled way to explain the tradeoff shown in Figure 6 (and illustrated in Figure 7). It decomposes the error into two interpretable gaps, which naturally leads to the observed non-monotonic behavior. We agree that deriving tighter bounds and explicitly quantifying each term would be valuable, and these can be important directions for future work.
> > >
> > > Thanks again for your helpful feedback.

---

### Official Review · Reviewer_QiWZ · 2026-03-21

**Soundness:** 3
**Presentation:** 3
**Significance:** 2
**Originality:** 3
**Overall Recommendation:** 4
**Confidence:** 4

**Summary:**

This work investigates how data size and data difficulty jointly influence supervised fine-tuning (SFT) performance. The authors show that, for a fixed data budget, performance exhibits a non-monotonic relationship with data difficulty: it improves as difficulty increases, reaches an optimal intermediate level, and then degrades when the data become too hard. Importantly, this optimal difficulty shifts toward more challenging data as the available training data increases.

The study explains the observed trends through the interplay between two factors: the in-distribution generalization gap and the extrapolation gap. Easy training data limit extrapolation to harder test cases, while overly difficult data hinder effective learning due to insufficient generalization. Increasing data size mitigates both gaps.

Empirically, the authors identify four key observations: (1) performance improves with data size before saturating; (2) difficulty has a non-monotonic effect with an interior optimum; (3) the optimal difficulty increases with data size; and (4) difficulty is relative to model capability, with stronger models benefiting from harder data. The authors further provide a learning-theoretic perspective to explain these behaviors.

**Compliance With Llm Reviewing Policy:**

Affirmed.

**Final Justification:**

After reviewing the authors' rebuttal and the new experimental results provided, I have decided to upgrade my rating based on the following:
1. The authors have exceptionally addressed my primary concerns regarding the robustness of the difficulty metric and the generalization of their findings. Specifically, the additional experiments using pass rate as difficulty metrics demonstrate that the observed phenomena are not artifacts of a single metric.
2. The extension of the 2D experiments to the Llama model family and science-oriented tasks strengthens the soundness of the paper.
3. The authors have committed to improving figure labels and formalizing definitions in the revised version, which addresses my concerns regarding clarity.
The rebuttal was comprehensive and provided strong empirical evidence that was missing in the initial submission.

**Key Questions For Authors:**

1. In Section 4, how does your formulation account for relative versus absolute data difficulty? Specifically, is difficulty defined independently of the model, or does it implicitly depend on model capability?
2. How do you practically measure relative difficulty given a specific model–dataset pair? For example, beyond CoT length, are there model-dependent metrics that could better capture relative difficulty?
3. The paper emphasizes data size and difficulty as primary drivers of SFT performance. Could you elaborate on why these factors are more critical than others discussed in prior work (e.g., data diversity, distribution mismatch, or curriculum design)? Under what conditions might those factors dominate?
4. Your study uses CoT length and number of operators as a proxy for difficulty. How sensitive are your conclusions to these choices? Have you evaluated whether alternative difficulty measures lead to similar observations, and if so, how consistent are the results?

**Limitations:**

yes

**Strengths And Weaknesses:**

Strengths
1. The paper proposes an interesting and timely perspective on the interplay between data difficulty and data size in SFT, moving beyond the common practice of uniformly selecting easy or hard data.
2. The observed non-monotonic relationship between difficulty and performance, as well as the shifting optimal difficulty with increasing data size, are clearly demonstrated. The performance curves (e.g., Figures 3 and 4) effectively illustrate these trends.
3. The interpretation based on the interaction between generalization and extrapolation gaps is conceptually appealing and provides a potentially useful lens for understanding SFT behavior.

Weaknesses
1. The paper is sometimes difficult to follow due to imprecise or delayed definitions. Key concepts—such as generalization gap and extrapolation gap—are introduced without sufficient formalization early on. Additionally, there are several grammatical issues that affect readability.
2. While the paper focuses on data size and difficulty, prior work has highlighted other important factors (e.g., data diversity, task distribution, and model capability). The paper would benefit from a clearer justification of why the interplay between data size and difficulty is prioritized over these other factors.
3. The notion of relative difficulty (with respect to model capability) is mentioned but not explicitly incorporated into the analysis. Moreover, the paper uses different difficulty metrics across sections (e.g., CoT token length vs. number of operations in experiments) without sufficient explanation or justification for this shift.
4. Although related work mentions multiple ways to measure difficulty, the paper does not adequately justify the choice of CoT length or discuss how sensitive the results are to this metric.
5. The experiments are somewhat limited in scope. Additional evaluations across diverse tasks and model scales would strengthen the claim that the observed interplay between data size and difficulty is generalizable.
6. Figures 3 and 4 lack clearly specified units or scales for performance, making it difficult to interpret the magnitude of improvements.
7. The related work section is relatively brief and does not sufficiently contextualize the proposed approach, particularly regarding difficulty metrics and the origin of the generalization–extrapolation framework.

---

> ### Author Rebuttal · Authors · 2026-03-31
>
> Thanks for the reviewer's careful reading of our paper and for providing detailed, constructive feedback. The reviewer's suggestion motivates us to significantly generalize our experiment setup to test our foundings. These results are consistent with existing observations and improve our work.
>
> 1. **Using pass rate and model likelihood as difficulty metric**. Based on Q2 and Q4 regarding difficulty metric, we added experiments using the pass rate as metric. The results are shown below, which are consistent with our current results. Further, we have also covered loss-based difficulty in Figure 10, Appendix B which also confirmed our findings.
>
> https://files.catbox.moe/bm3867.pdf
>
> **Rebuttal Figure 1**: 2D performance improvement over base model Qwen2.5-Math-1.5B, trained on OpenMath, evaluated by average accuracy on Math500, AIME24, and Minerva Math. Difficulty is measured by (1.0 - pass rate).
>
> 2. **Extension to science tasks and another model family**. By the suggestion on generalization in W5, we added experiments on science task and Llama model respectively. The results are shown below. Model performance exhibits similar trends as main experiments.
>
> https://files.catbox.moe/uba5m5.pdf
>
> **Rebuttal Figure 2**: 2D improvement over base model Qwen2.5-Math-1.5B, trained on OpenScience, evaluated by average accuracy on MMLU and GPQA. Difficulty is measured by CoT length.
>
> https://files.catbox.moe/svb93a.pdf
>
> **Rebuttal Figure 3**: 2D improvement over base model Llama-3.2-3B, trained on OpenMath, evaluated by average accuracy on Math500, AIME24, and Minerva Math. Difficulty is measured by CoT length.
>
> Below, we discuss each  comments and questions in more details.
> ## Response to Q1&Q2&Q4&W3&W4(Selection of Difficulty Metric)
> >Q1. Is difficulty defined independently of the model, or does it implicitly depend on model capability?
>
> We appreciate this important question. In our work, difficulty is **model-dependent** and defined in a common-sense, natural way. It measures how difficult the **question** is for the model as well as how easy it is to learn the **solution**.
>
> >Q2. How do you practically measure relative difficulty given a specific model–dataset pair? For example, beyond CoT length, are there model-dependent metrics that could better capture relative difficulty?
>
> > Q4. Have you evaluated whether alternative difficulty measures lead to similar observations, and if so, how consistent are the results?
>
> Thank you for this insightful concern. In prior works no universal metric exists for difficulty. We adopt three widely accepted metrics: **CoT length**, **NLL loss** and **pass rate**. (In the iGSM setup, number of operations is equivalent to CoT length). We think all three metrics capture different aspects of data difficulty, therefore we don't claim any of them is better.
>
> The results all exhibit similar trends(Section 3, Figure 3,4; Appendix B, Figure 10; Rebuttal Figure 1) : an optimal difficulty level increasing with data size. This consistency suggests that **our conclusions reflect intrinsic effect of data difficulty rather than an artifact of any particular metric**. We will highlight these three results in the main text.
> ## Response to W5(Generalization Across Tasks and Models)
> To check whether our findings generalize across tasks and model families, we conduct two separate sets of 2D experiments, one on science task and one on Llama3.2-3B model. Results are shown in the link of Rebuttal Figure 2 and 3 respectively, which align well with our current findings.
>
> Regarding model size, extending the full 2D experiment to 14B or larger models is not feasible due to computational constraints. However, we already compared 1.5B and 7B models in Figure 3(Section 3). Both model sizes exhibit the same core phenomena, which supports the generality of our findings.
>
> We will add these experiments into the revised version.
> ## Response to Q3&W2(Interplay With Other Factors)
> >Q3. Could you elaborate on why these factors are more critical than others discussed in prior work? Under what conditions might those factors dominate?
>
> Thank you for this informative concern. We fully agree all those factors are important, and they interact in complex ways. We do not claim data size and difficulty are the most critical factors. Instead, our framework studies the interplay between data difficulty and data size, while ****holding other variables fixed****(Controlled by sampling from the same dataset in real-world setting and generating data from the same format in iGSM).  Studying how these factors interact with others is a promising extension in future work, we will add this discussion into the future work section.
> ## Response to W1&W6&W7 (Figure Clarity, Concept Formalization and Related Work)
> Thank you for raising up. For Figures 3 and 4, all metrics are reported in averaged percentage accuracy(%). We will add explicit axis labels in the revision. We will introduce the concepts earlier and expand related work accordingly.

---

> > ### Author Rebuttal · Reviewer_QiWZ · 2026-04-03
> >
> > Thank you for your detailed rebuttal and for the significant effort you put into conducting the additional experiments in such a short timeframe.
> > The new results -  the evaluations using alternative difficulty metrics and the extension to the Llama model family and science tasks, effectively address my concerns regarding the robustness and generalization of your findings. The consistency across these different setups strengthens the paper's core claims.
> > I also appreciate your clarifications regarding the interplay with other factors and the commitment to improving the figure clarity and conceptual formalization in the revised version.
> > With all these comprehensive responses and the new evidence provided, I have increased my score to reflect.

---

> > > ### Author Response · Authors · 2026-04-04
> > >
> > > Thank you for raising your score and for the encouraging acknowledgement. We are glad that the additional experiments and clarifications have addressed your concerns. Your constructive feedback has significantly improved the quality of our work, and we will carefully incorporate all the suggested revisions—including clearer figure labels, earlier concept formalization, and an expanded related work section—into the final version.
> > >
> > > Thanks again for your time and effort in reviewing our paper.

---

### Decision · Program_Chairs · 2026-04-30

**Decision:**

Accept (regular)

**Comment:**

This paper investigates the joint influence of data difficulty and dataset size on the supervised fine-tuning performance of large language models. The authors challenge the common heuristic of uniformly filtering for either easy or hard data, demonstrating instead that the optimal data difficulty is dependent on the dataset size. However, several shared concerns were raised during the initial review phase, such as the concern of generalizability, metrics used, and theoretical and practical utility. The authors' rebuttal resolved the most critical concerns regarding the robustness and generalizability of their empirical findings. I feel that the paper's core empirical insights are valuable, and therefore suggest a weak acceptance of it.